# A two-kinesin mechanism controls neurogenesis in the developing brain

Paige Helmer [1,2✉] & Richard B. Vallee[1,2✉]

During the course of brain development, Radial Glial Progenitor (RGP) cells give rise to most of the neurons required for a functional cortex. RGPs can undergo symmetric divisions, which result in RGP duplication, or asymmetric divisions, which result in one RGP as well as one to four neurons. The control of this balance is not fully understood, but must be closely regulated to produce the cells required for a functioning cortex, and to maintain the stem cell pool. In this study, we show that the balance between symmetric and asymmetric RGP divisions is in part regulated by the actions of two kinesins, Kif1A and Kif13B, which we find have opposing roles in neurogenesis through their action on the mitotic spindle in dividing RGPs. We find that Kif1A promotes neurogenesis, whereas Kif13B promotes symmetric, non-neurogenic divisions. Interestingly, the two kinesins are closely related in structure, and members of the same kinesin-3 subfamily, thus their opposing effects on spindle orientation appear to represent a novel mechanism for the regulation of neurogenesis.

[1] Department of Pathology and Cell Biology, Columbia University Medical Center, New York, NY 10032, USA. [2] Department of Biological Sciences, Columbia University, New York, NY 10032, USA. ✉email: ph2497@columbia.edu; rv2025@cumc.columbia.edu

Mammalian brain development is a highly complex process involving multiple mechanisms for controlling neuronal differentiation, distribution and number. Microtubule motor proteins have been found by us and others to contribute to radial glial progenitor (RGP) proliferation and neuronal migration[1,2]. However, relatively little is known about potential roles of the motor proteins in specifying neuronal number and brain size.

In the course of higher eukaryotic brain development, neural progenitor cells undergo numerous rounds of mitosis, generating the large number of neurons and glia that will populate the mature brain. Neurogenesis is controlled by the balance of symmetric vs. asymmetric divisions by neural stem cells, known as radial glial progenitors (RGPs; Fig. 1)[3]. During early development, most RGP divisions are symmetric, increasing their number. The percentage of asymmetric divisions then gradually increases (Fig. 1a). This allows the size of the RGP cell pool to be maintained, while also generating a sufficient number of neurons to populate the expanding neocortex.

In mammals, neurogenesis can occur directly from RGPs, or indirectly through intermediate progenitors (IPs)[3]. Direct neurogenesis yields a single RGP cell and a neuron. Indirect neurogenesis can yield two or more neurons as a result of further IP cell divisions (Fig. 1)[4]. The mechanisms responsible for controlling division symmetry, IP production, and neuronal number are only partially understood.

Kif1A is a member of the microtubule-based kinesin motor protein family, which has diverse roles in intracellular transport of vesicular and nuclear cargoes. We have found that during brain development Kif1a is first involved in the cell proliferation behavior known as interkinetic nuclear migration (INM) a key feature of neural progenitor development[1]. During INM the nucleus exhibits striking oscillatory behavior, first traveling basally away from the ventricle during G1, and then apically back to the ventricle during G2, though the purpose of this behavior remains incompletely understood. Kif1A then further contributes to post-mitotic neuronal migration toward the developing cortex[1, 2]. Kif1A is also well-known for its more general roles in the transport of dense-core vesicles, including those involved in secretion of BDNF in post-mitotic neurons, which is required for proper neuronal migration[5]. We have also noted that Kif1a knockdown reduces the ratio of asymmetric:symmetric divisions in RGPs[5]. This outcome was correlated with an increase in the angle between the RGP cell division axis and the ventricular surface (VS) of the developing brain[5]. Such an association is consistent with extensive literature supporting a relationship between mitotic orientation and neurogenesis[6], and suggests a role for Kif1A in such a mechanism.

A related, but clearly distinct, kinesin motor protein in *Drosophila*, khc73 (Kif13B in vertebrates) has been implicated in the control of mitotic spindle orientation in neuroblasts[7,8]. RNAi for khc73 in these cells resulted in an increased angle between the mitotic spindle and polarity factors at the cell membrane[8]. This result is opposite to the effect we observed from reduced Kif1A expression in embryonic rat brain[5].

Mammalian Kif13B has no known role in brain development, though it has been implicated in cholesterol metabolism, axon specification, and myelination following nerve injury[9–11]. A role for Kif13B in mammalian brain development, however, remains unexplored.

Kif1A and Kif13B are strikingly similar in overall structure, and the motor domains of the two kinesins are particularly close in primary structure[12,13]. The two forms of kinesin also exhibit some similarities in domain organization (Fig. 1b), with the noteworthy exception of the tail domains. Kif1A contains a pleckstrin homology (PH) domain, whereas Kif13B contains a CAP-Gly domain for MT plus end binding, a surprising and unique feature among all known kinesins.

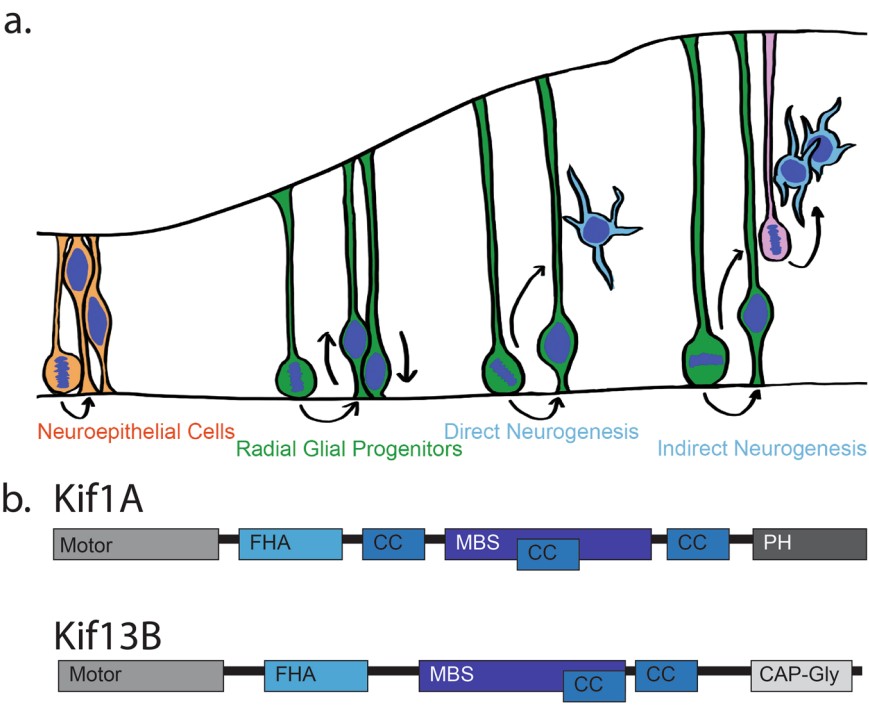

**Fig. 1 Brain development and kinesins. a** Diagrammatic representation of rodent brain cortical development. Neuroepithelial cells (orange) divide symmetrically, increasing their numbers. These cells develop into radial glial progenitor cells (RGPs, green). These cells divide symmetrically to expand their pool, or asymmetrically to produce neurons (blue). Neurogenesis can occur directly, or indirectly through intermediate progenitors (IPs, pink). **b** Domain maps of Kif1A and Kif13B, showing their structural similarities and differences, notably, the presence of a lipid binding PH domain in Kif1A and a microtubule plus-end binding CAP-Gly domain in Kif13B.

The current study was initiated to test for a possible role for Kif13B in mammalian brain development and the relative roles of the two related kinesins in this process, and, in particular, RGP cell fate. We find that Kif13B has a previously undescribed and important role in mammalian neurogenesis, complementary to, but clearly distinct from and in important regards, opposite to, those for Kif1A. Our results suggest that Kif1A and Kif13B might, indeed, serve together in an important mechanism for controlling the balance between symmetric and asymmetric divisions, and, in turn, the timing and extent of neurogenesis in the higher eukaryotic brain.

## Results

**Reduced Kif13b expression increases the ratio of neurons to progenitors in the developing rat brain.** To compare the roles of Kif13B vs. Kif1A in rat brain cortical development, we injected embryonic day 16 (E16) rat embryos in utero with an empty GFP control plasmid, or shRNAs designed for knockdown of Kif13b or Kif1a. Brains were fixed at a range of time points from E18 to E20, sectioned, and then imaged to determine the effects of reduced Kif13b or Kif1a expression at various developmental stages.

The cortex at E18-E20 consists of four regions: the cortical plate (CP), intermediate zone (IZ), sub-ventricular zone (SVZ) and ventricular zone (VZ). By E19, Kif13b shRNA had caused a striking increase in the number of transfected cells located within the cortical plate (CP): 36% ± 10% of transfected cells reached the CP by E19, compared to 15% ± 6% in control brains (Fig. 2b, e). This is a remarkable result, directly opposite to the decrease in neurogenesis we have observed in Kif1a shRNA-expressing brains (Fig. 2b, e)[1,5].

As early as E18, Kif13b knockdown had also caused a clear increase in the number of transfected cells reaching the IZ and SVZ, with 71% ± 8% of shRNA-transfected cells found in those regions compared to 61% ± 7% in control brain (Fig. 2a, d). This indicates that during the period from E16 to E18, Kif13B knockdown causes a surprising increase in neurogenesis. Curiously, by E20, there was no significant difference in the percent of cells in the CP under Kif13b shRNA vs. to GFP control conditions (Fig. 2c, f). This indicates that, although reduced Kif13B expression results in an increase in early neurogenesis as seen by E18, this effect may not be sustained. Interestingly, if the cortical plate at E20 is divided into equal-size upper and lower bins, the control brains showed roughly equal cell distributions of cells in the two regions. Kif13b shRNA, in contrast, resulted in a clear accumulation of cells in the upper bin, closer to the pial surface of the developing cortex (Fig. 3). When rat embryos are electroporated at E16 and analyzed at E20, the transfected cells in the lower cortical region are newer neurons in the process of outward migration, and the transfected cells closer to the pial surface are older neurons that have already migrated. Reduction in the number of Kif13b shRNA-expressing neurons in the lower CP region indicates that by E20, fewer new neurons may be produced with Kif13b shRNA compared to control brains.

**Expression of Kif1A vs. Kif13B controls RGP cell fate.** To better understand the opposing effects of Kif1a and Kif13b knockdown on neurogenesis, we used a variety of histological markers for cell types in the VZ and SVZ. Phospho-histone-H3 (PH3) was used to detect actively dividing cells. RGPs divide exclusively at the VS, due to the cell-cycle dependent nature of INM. Intermediate progenitors (IPs) however, do not contact the VS while dividing, and are located at the basal edge of the VZ and in the SVZ. By observing the location of GFP+/PH3+ cells in relation to the

ventricular surface, the ratio of apical progenitors (APs) to IPs can be calculated.

In control E19 brain, 75% ± 12% of GFP/PH3+ RGPs were seen dividing at the ventricular surface. In contrast, 25% ± 12% of GFP/PH3+ cells were found dividing more basally (IPs) (Fig. 4a). In Kif13b shRNA conditions, however, only 55% ± 3% of GFP/PH3+ cells were located at the ventricular surface, with 45% ± 3% of GFP/PH3+ cells dividing more basally. These results agree with our previous observations of increased neurogenesis under Kif13b shRNA conditions (Fig. 2), as IP-mediated neurogenesis produces more neurons from a single RGP division than direct neurogenesis from APs alone.

Kif1a shRNA-expressing brains exhibited a marked reduction in the percent of IPs as compared to the GFP-expressing control brain. Only 10% ± 2% of GFP/PH3+ cells with Kif1a shRNA treatment were observed dividing in the upper VZ and SVZ (Fig. 4a). This indicates a reduction in IP production, which would, in turn, result in a decrease in neurogenesis. This result agrees with the results we observed when viewing the percent of cells in the CP under Kif1a shRNA conditions (Fig. 2a).

During development, RGPs typically express Pax6, whereas IPs express TBR2. Occasionally, differentiating IPs express the two markers simultaneously, but most often cells will express Pax6, then TBR2 in sequence, with limited overlap[14]. To determine how RGP cell fate is affected under Kif1a vs. Kif13b knockdown conditions, we stained for Pax6, an RGP specific marker. Kif13b shRNA caused a marked decrease in GFP+/Pax6+ cells, from 84% ± 9% in brains expressing GFP control to 59% ± 4% following Kif13b shRNA (Fig. 4c). No difference in the percent of GFP+/Pax6+ cells was seen in response to Kif1a shRNA (Fig. 4c). These results indicate that Kif13b knockdown results in fewer cells maintaining RGP cell fate.

We also examined the effect of reduced Kif1a and Kif13b expression on the number of intermediate progenitors using an antibody to the IP marker, TBR2. The percent of GFP+/TBR2+ cells increased to 44% ± 12% in cells expressing Kif13b knockdown compared to 24% ± 5% of GFP-expressing control cells, indicating a marked increase in the number of IPs under Kif13b shRNA treatment compared to GFP control (Fig. 4e). Interestingly, the fraction of GFP+/TBR2+ cells was also increased by Kif1a shRNA, with 46% ± 13% of cells positive for GFP and TBR2 (Fig. 4e). As the percent of GFP+/Pax6+ cells is not reduced by Kif1a shRNA treatment, many of these cells are likely to be positive for both Pax6 and TBR2. Cells that express both of these markers are usually differentiating IPs[14]. This was confirmed by double staining for Pax6 and TBR2 (Fig. S1). GFP control brains showed 7% ± 1% of GFP+ cells that were positive for both Pax6 and TBR2. This number was increased to 13% ± 4% in Kif1a shRNA-expressing brains (Fig. S1). The increased number of these Pax6+/TBR2+ cells under Kif1a shRNA treatment indicates that without Kif1a expression, RGPs produce differentiating IPs rather than proliferative IPs that will divide to produce multiple neurons. Therefore, Kif1a shRNA results in fewer neurons per RGP division. This result agrees with our previous finding that fewer PH3+ cells are present in the upper VZ and SVZ following Kif1a shRNA expression.

**Kif1A and Kif13B knockdown have opposing effects on mitotic spindle orientation in dividing RGPs.** A number of variables have been proposed to play a role in RGP cell fate determination. These include inheritance of apical polarity factors such as adherens junctions, and exposure to Notch or Sonic Hedgehog signaling pathways, which can each be affected by mitotic spindle orientation[6,15–18]. We previously reported that reduced Kif1a expression decreases neurogenesis in embryonic rat brain[5]. This

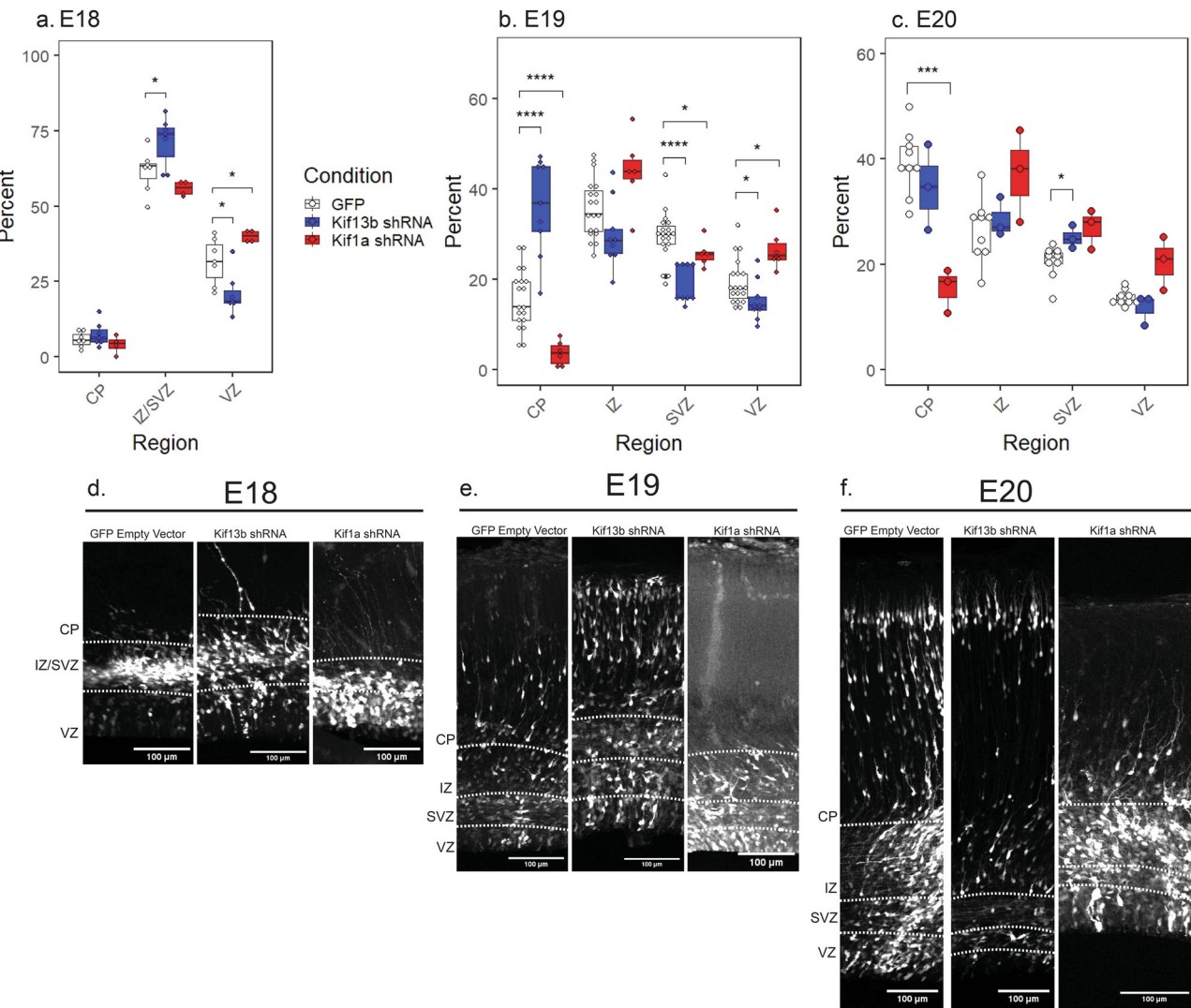

**Fig. 2 Kif13b and Kif1a shRNA have opposing effects on neurogenesis.** Percent of cells in CP, IZ, SVZ, or VZ of the cortex at (**a**) E18, (**b**) E19, and (**c**) E20. At E18, Intermediate zone and sub-ventricular zone are combined into one region (IZ/SVZ). Data plotted as interquartile range with 5–95% whisker range. (****$p < 0.0001$, ***$p < 0.001$, *$p < 0.05$. Analyzed using two sided unpaired $t$-test. Total GFP control brains: E18 = 7, E19 = 19, E20 = 3. Total Kif13b shRNA brains: E18 = 7, E19 = 9, E20 = 9. Total Kif1a shRNA brains: E18 = 4, E19 = 6, E20 = 3. At least 118 cells from each brain were included). Representative images of GFP control, Kif13b shRNA, and Kif1a shRNA-expressing rat brains at (**d**) E18, (**e**) E19, and (**f**) E20. White dashed line indicates the border between cortical regions. Scale bar = 100 μm.

effect was correlated with an increase in the angle between the cytokinetic plane and the VS in RGPs as monitored by live imaging of brain slices. The cytokinetic plane, measured previously in live brain slices, and the orientation of the mitotic spindle, measured here, are roughly 90° apart from each other. Spindle orientation in fixed tissue is measured by calculating the angle between the centrosomes and the VS, a measurement that cannot be done in live cells that are not expressing a centrosomal marker. Thus, an increase in cytokinetic plane corresponds to a decrease in mitotic spindle orientation.

We also note that RNAi of the Kif13b orthologue, khc-73, in *Drosophila* neuroblasts showed a role for this kinesin in spindle orientation[7,8]. However, lack of khc-73 expression resulted in an increase in the spindle angle relative to polarity factors at the cell membrane, which is the opposite of the effect Kif1a shRNA has on RGP division plane. Given the opposing effects of Kif1A and Kif13B on neurogenesis revealed in the current and earlier studies, and the potential for each kinesin to affect mitotic spindle orientation, we directly compared this behavior in embryonic rat brain under Kif1a and Kif13b shRNA conditions.

To evaluate spindle angle, we stained E19 embryonic rat brains for PH3 as a marker for mitotic chromosomes, and centrin-3 or γ-tubulin as markers for centrosomes and mitotic spindle poles (Fig. 5). Kif1a shRNA resulted in a reduction of RGP spindle angle, from a mean of 23.1° ± 19.4° with GFP control to 15.2° ± 16.4° with Kif1a shRNA, as we expected from previous studies (Fig. 5a–c)[5].

Over the course of neural development, the mitotic spindle of RGPs remains nearly horizontal during early development, and spindle angle increases over time as more neurons are produced[6]. At E19, for example, the average spindle angle is already somewhat randomized, and an increase in spindle angle is more difficult to distinguish. Because of this phenomenon, observing spindle orientation at E19 in rat brain does not provide an accurate comparison for Kif13b shRNA effects on spindle orientation, and an earlier stage of development is required. For this reason, we measured mitotic spindles at E18 instead (Fig. 5d–f). As predicted, at E18 expression of Kif13b shRNA increased the angle of the mitotic spindle in dividing RGPs, from a mean of 14.6° ± 8.5° to 21° ± 16.7° with Kif13b shRNA (Fig. 5d–f).

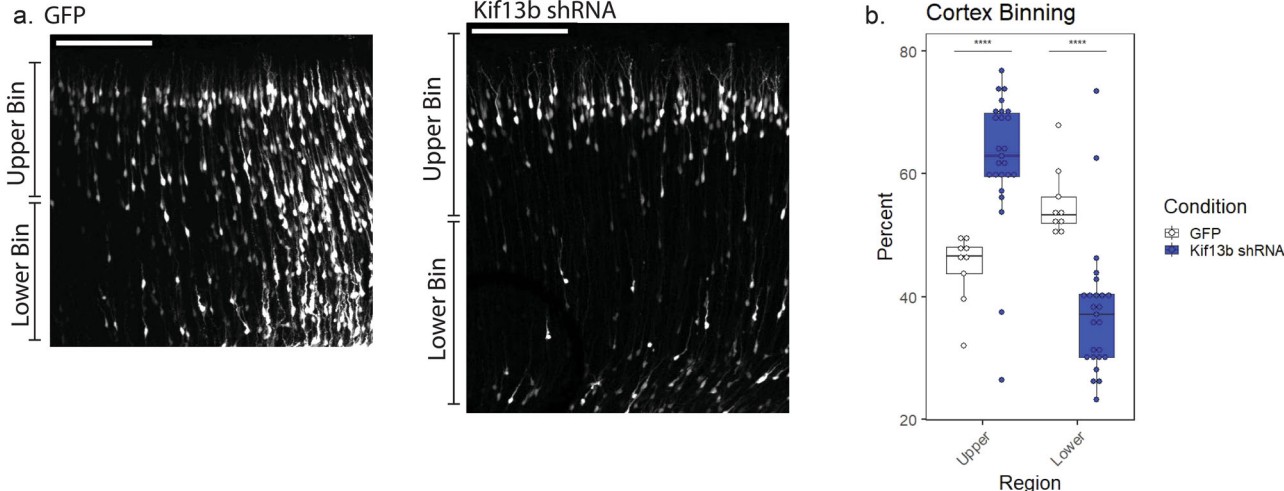

**Fig. 3 Kif13b shRNA disrupts the distribution of neurons within the cortical plate. a** Representative images of the cortical plate in E20 rat brain expressing GFP control or Kif13b shRNA. Scale bar = 100 μm. **b** Distribution of GFP or Kif13b shRNA-expressing rat brain in E20 brain. The percent of cells in each of two equal-sized horizontal bins was determined. Data plotted as interquartile range with 5–95% whisker range. (****$p < 0.0001$. Analyzed using two sided unpaired $t$-test. Total GFP brains = 3, Kif13b shRNA brains = 9. At least 142 cells from each brain were included).

While the change in mitotic spindle orientation is clear in rat brain at E18, analysis of earlier stages of development cannot be done using this model. Most embryos do not survive IUE procedures performed prior to E16, and using shRNA requires at least 48 h for effective knockdown of target genes. Thus, E18 is the earliest stage we can observe the effects of Kif13b shRNA in the rat brain. In order to observe effects of Kif13b at earlier developmental stages, we made use of a mouse model for conditional Kif13b deletion[11]. By mating C57B/6 mice with LoxP sites introduced in the Kif13b gene with mice expressing Cre recombinase under control of the Nestin promotor, we were able to selectively remove expression of Kif13b in RGPs[11, 19]. This model allows for observation of Kif13B effects on spindle orientation in RGPs at earlier stages than is possible in rat brain using IUE.

By E14.5, RGPs showed an increase in mitotic spindle angle in Kif13b cKO mice compared to control mice (Fig. S2). WT mice had a mean spindle angle of 13.6° ± 13.1°. In contrast, the Kif13b cKO mice had a mean spindle angle of 16.5° ± 14.8° (Fig. S2a, b). Although the difference in the mean spindle angle is small at E14.5, the distribution of spindle angles is quite different. In WT mice, only roughly 8% of dividing cells have a spindle angle greater than 30°, while close to 18% of dividing RGPs in the Kif13B cKO show spindle angles 30° or greater. The larger percentage of RGPs with more extreme spindle angles in the Kif13b cKO mouse will produce more IPs, which will produce at least two neurons per RGP division. The increase in overall spindle angle at E14.5 in Kif13b cKO mice indicates that more neurogenic divisions occur in the absence of Kif13B. In addition to measuring the spindle angle in E14.5 mouse brains, we also stained for Pax6, TBR1, and PH3 in order to observe overall effects on cortical development in the Kif13b cKO condition. TBR1, a marker for newly produced neurons, was chosen in order to determine if neuron production was increased in the Kif13b cKO mouse model at an early cortical developmental stage, when there are relatively few TBR1+ cells. As expected from the results seen in rat brain, Kif13b cKO resulted in a decrease in the number of Pax6 positive cells (Fig. S2d), and an increase in the number of TBR1 positive cells (Fig. S2e). Kif13b cKO mice also showed an increase in the percentage of PH3 positive cells that were located at a distance from the ventricular surface, indicative of proliferating IPs (Fig. S2f). These results, together with the

results seen in rat brains at E18 and E19, show that loss of Kif13b in RGPs results in increased mitotic spindle angles, which in turn increases the number of IPs produced and increases neurogenesis early in development.

After observing the effects of Kif1a shRNA and Kif13b shRNA independently, we next combined the two. By adding the Kif13b shRNA sequence into a vector with RFP rather than GFP, we were able to include both constructs for IUE and observe the effects of loss of expression of both kinesins. As might be expected from the opposing effects of each individual shRNA, the combination of shRNAs results, in most cases, in a rescue of the phenotypes seen in each individual condition (Fig. S3). The percent of transfected cells positive for TBR2 is unchanged, as well as the percent of transfected cells in most regions of the cortex (Fig. S3c, e). There is a slight, but significant, reduction in the percent of Pax6+ cells with the double shRNA knockdown (Fig. S3a), as well as a slight reduction in the percent of cells in the VZ (Fig. S3e).

We next wanted to observe if the overexpression of each kinesin resulted in an opposite phenotype from the corresponding knockdown, which could indicate that the levels of Kif1A and Kif13B present are regulated in order to control spindle orientation, or if another regulation mechanism controls the effect of each kinesin during development. Kif1A and Kif13B were each cloned into a vector over-expressing the full-length kinesin, as well as diffuse RFP throughout the cell via an internal ribosomal entry site (IRES). Both Kif1A and Kif13B overexpression (OE) showed little effect on Pax6 expression (Fig. S4a, b) or TBR2 expression (Fig. S4c, d). There was a slight, but significant increase in the percent of cells in the VZ in the Kif1A OE condition, from 19.33% ± 4.96% in control brains to 30.29% ± 8.59% (Fig. S4e, f). This overall lack of phenotype in Kif13B and Kif1A OE conditions indicates that the levels of each kinesin is not the main regulatory mechanism of RGP cell fate, but rather that there may be an additional regulatory mechanism to control when Kif1A and Kif13B function to affect spindle orientation.

We next attempted to determine the effects of the double KD and overexpression of each kinesin on mitotic spindle. However, all three conditions significantly reduced the mitotic index of the RGP cells (Fig. S5a, b). This drastic reduction of dividing cells made analysis of spindle orientation and the precent of IPs

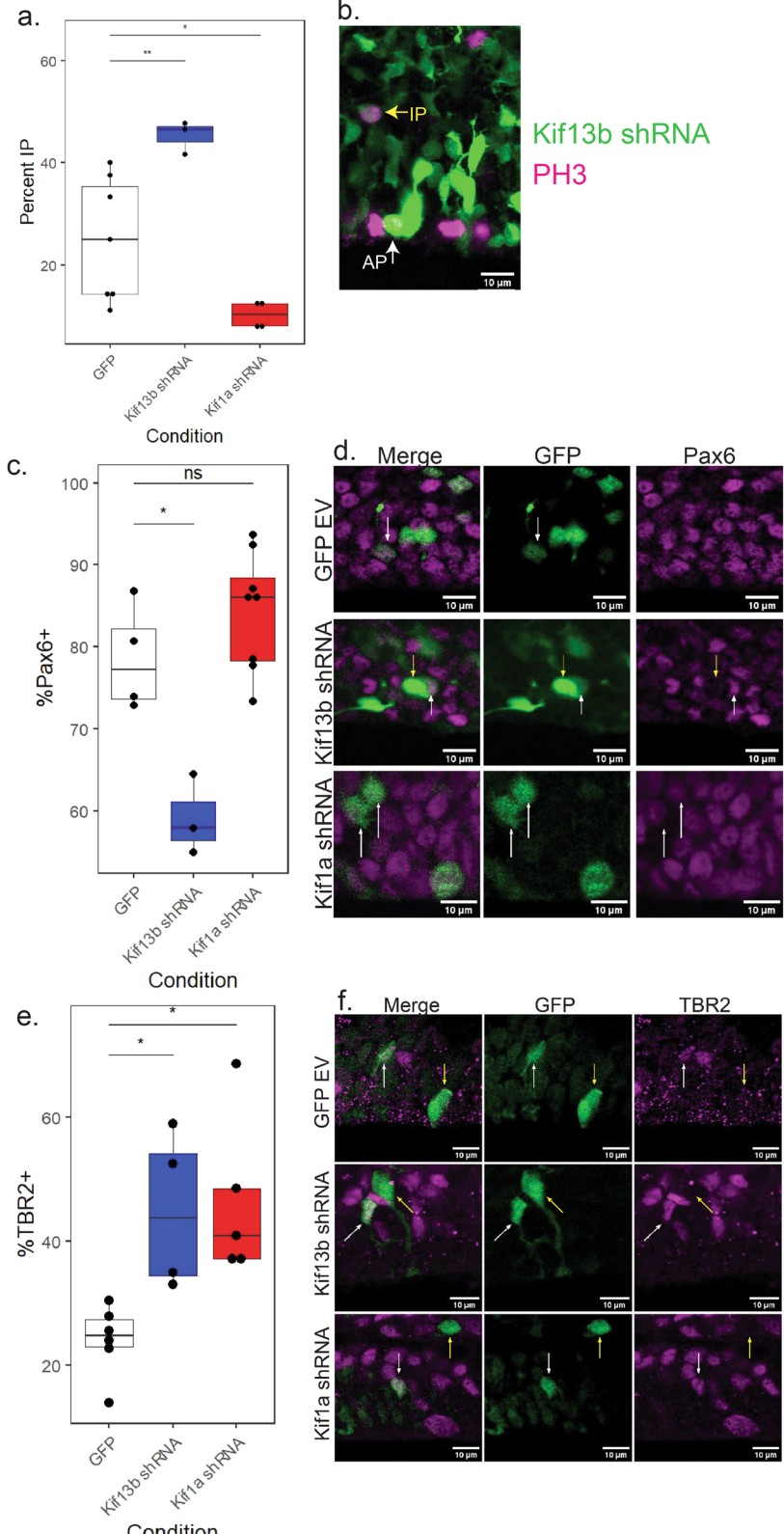

present in each condition ineffective. As Kif1A is the motor responsible for transporting the RGP nucleus basally during INM[1] and knockdown of Kif1A results in RGPs that are accumulated close to the ventricular surface[1,5], the distance from the RGP cell body to the VS was measured. We reasoned that overexpression of the basally directed motor might increase the RGP distance to the VS, which is indeed what we saw in the

Kif1A OE condition (Fig. S5c, d). With cells artificially recruited away from the ventricular surface, this could explain the reduction in mitotic index, as RGPs need to reach the ventricular surface in order to divide. We also measured the RGP distance to the VS in the double KD and Kif13B OE conditions to see if a similar effect was seen that might explain the reduced mitotic index (Fig. S5c, d). Interestingly, while Kif1a shRNA reduces the

**Fig. 4 Kif13b and Kif1a shRNA have opposite effects on IP cell number. a** Quantification of PH3 positive cells dividing away from the ventricular surface. Intermediate progenitor (IP) percentage was calculated as number of GFP+/PH3+ cells observed with no contact with the ventricular surface divided by total number of GFP+/PH3+ cells. Data plotted as interquartile range with 5–95% whisker range. (**$p < 0.01$, *$p < 0.05$. Analyzed using two sided unpaired $t$-test. Total brains for GFP control = 8, Kif13b shRNA = 4, Kif1a shRNA = 4. At least 21 PH3+ cells from each brain were included).
**b** Representative image showing an IP cell (yellow arrow) and an AP cell (white arrow) in a Kif13b shRNA-expressing E19 brain. Scale bar = 10 μm.
**c** Quantification of Pax6+ cells in E19 rat brain VZ. The percentage of Pax6+ cells was calculated as number of GFP+/Pax6+ cells divided by total GFP+ cells at the ventricular surface. Data plotted as interquartile range with 5–95% whisker range. (**$p < 0.01$, Analyzed using two sided unpaired $t$-test. Total brains for GFP control = 5, Kif13b shRNA = 3, Kif1a shRNA = 9. At least 81 cells from each brain were included). **d** Representative images of GFP, Kif13b shRNA, or Kif1a shRNA expression (green) and Pax6 staining (magenta). White arrows indicate GFP+/Pax6+ cells and yellow arrows indicate GFP +/Pax6− cells. Scale bar = 10 μm. **e** Quantification of TBR2+ cells in E19 rat brain VZ. Percentage of TBR2+ cells was calculated as number of GFP +/TBR2+ cells divided by total GFP+ cells at the ventricular surface. Data plotted as interquartile range with 5–95% whisker range. (*$p < 0.05$. Analyzed using two sided unpaired $t$-test. Total brains for GFP control = 6, Kif13b shRNA = 5, Kif1a shRNA = 5. At least 52 cells per brain were included).
**f** Representative images GFP, Kif13b, or Kif1a shRNA (green) and TBR2 (magenta). White arrows indicate GFP+/TBR2+ cells and yellow arrows indicate GFP+/TBR2− cells. Scale bar = 10 μm.

RGP distance to the VS, Kif1a shRNA in combination with Kif13b shRNA increases this distance. Given that Kif13B OE has no effect on INM (Fig. S5c, d), it is unclear why this effect is seen, but, similar to the Kif1A OE condition, this effect could explain the reduction of mitotic index in the double KD condition, as the cells are unable to effectively reach the VS in order to divide. It is unclear what mechanism by which Kif13B OE reduces mitotic index, and it may be related to Kif13B transport of other cargoes within RGPs, such as Dlg1, which is involved in cytokinesis[11,20], early endosomes[21], Rab6 vesicles[22], or PIP3 vesicles[23], among other cargoes. Kif13B also functions in ciliary length and signaling, which could be affected by overexpression of Kif13B, and affect the ability of cells to go through mitosis[24,25].

## Discussion

**Kif1A and Kif13B have opposing roles in the control of neurogenesis in the developing brain.** Our previous work revealed that decreased Kif1A expression in RGPs results in a dramatic reduction in neurogenesis[1,5]. In marked contrast, we have now found that decreased Kif13b expression a dramatically increases in neurogenesis at the same developmental stage. This finding indicates that these two structurally-related, though distinct forms of kinesin have opposing roles in overall brain development, and must, in some way, coordinate cell fate during neurogenesis.

**Kif1A and Kif13B control rat brain neurogenesis via effects on in rat RGP mitotic spindle orientation.** Our results reveal that the control of neurogenesis by Kif1A and Kif13B begins at the RGP cell stage. By using cell-specific markers, it was apparent that knockdown of Kif1a or Kif13b results in specific changes in the developmental stage of cells present in the VZ and SVZ. Kif1a shRNA reduced production of IPs, as evidenced by a high percentage of Pax6+ cells, and a reduction in PH3+ cells in the upper VZ and SVZ. This reduction in IPs was also correlated with an increased percentage of Kif1AshRNA-expressing cells remaining in the VZ and SVZ.

Kif13b shRNA resulted in the opposite effect: an increase in IP production. This outcome is reflected in an increase in the number of PH3+ IPs seen in the upper VZ and SVZ. As expected from these findings, but remarkable nonetheless, the increase in IPs ultimately resulted in an increased number of neurons in the cortical plate under Kif13b shRNA conditions at E19. Given that at E20, the percent of cells in the cortical plate no longer differs between control brains and Kif13b shRNA, this may indicate that Kif13B plays an important role in maintaining the RGP cell population through development. This is supported by the observation that the cells in the CP with Kif13b shRNA are located near the pial surface of the developing brain in E20 rat

brain, with very few cells appearing in the apical half of the CP. As new neurons migrate from the VS outward past older neurons toward the pial surface of the brain, the sparsity of cells in the deeper region of the CP at E20 after transfection at E16 supports the conclusion that there is a decrease in the number of neurons being produced by E20 under Kif13b shRNA conditions.

**Kif1A and Kif13B have opposing roles in spindle orientation in RGPs.** As previously shown, reduction of Kif1A expression in the developing cortex increases the frequency of symmetric RGP divisions, which was correlated with an increase in the angle of the RGP cell division plane[5]. In this study, we show that this increase in division plane is confirmed by a decrease in spindle angle relative to the ventricular surface. In contrast, Kif13b shRNA resulted in an increase in spindle angle in E18 rat brain. This increase in spindle angle is more dramatic in Kif13b cKO E14.5 mouse brain, which corresponds to E16 in rat. The opposing effects on RGP spindle we observe to result from Kif1a and Kif13b shRNA in rats or deletion in mice seem quite remarkable, given the similarity in Kif1A and Kif13B sequence and structure.

An earlier hypothesis regarding control of RGP cell fate in zebrafish involved the relative exposure of RGP nuclei to a Notch gradient during INM. In this model, nuclei that remain close to the VS during INM are exposed to high levels of Notch and, therefore, are more likely to undergo symmetric divisions. Nuclei that travel farther from the ventricular surface and away from the higher Notch levels would be more likely to result in an asymmetric division. Intriguingly, given that Kif13B does not affect INM and still has a remarkable effect on neurogenesis, this theory cannot explain the effects Kif13B has on IP production and neurogenesis.

Kif1A and Kif13B are very closely related in domain organization. However, they have distinct roles in RGP cell fate and overall brain and structure and function. Kif1A serves to promote neurogenic divisions by increasing randomization of the mitotic spindle orientation. Reduction in Kif1A levels results in fewer asymmetric divisions, fewer IPs, and fewer neurons in the developing brain (Fig. 6). Kif13B plays a directly opposing role to Kif1A, and serves to stabilize spindle orientation. Loss of Kif13B results in an increase in the percentage of asymmetric RGP divisions early in development, more IPs, and increased early neurogenesis (Fig. 6).

How the two kinesins, with a common microtubule plus end-directed motor domain, could achieve distinct effects on mitotic spindle orientation is a fascinating issue. The answer might involve differential interactions of the two kinesins with mitotic spindle MTs and components of the mitotic cell cortex. Kif13B has been confirmed to interact with Dlg1 via its MBS domain[7,26]. Kif1A and Kif13B both contain MBS domains of similar structures[27], which could indicate that Kif1A might also interact with Dlg1, and indeed Kif1A fragments have shown interaction with Dlg proteins in brain

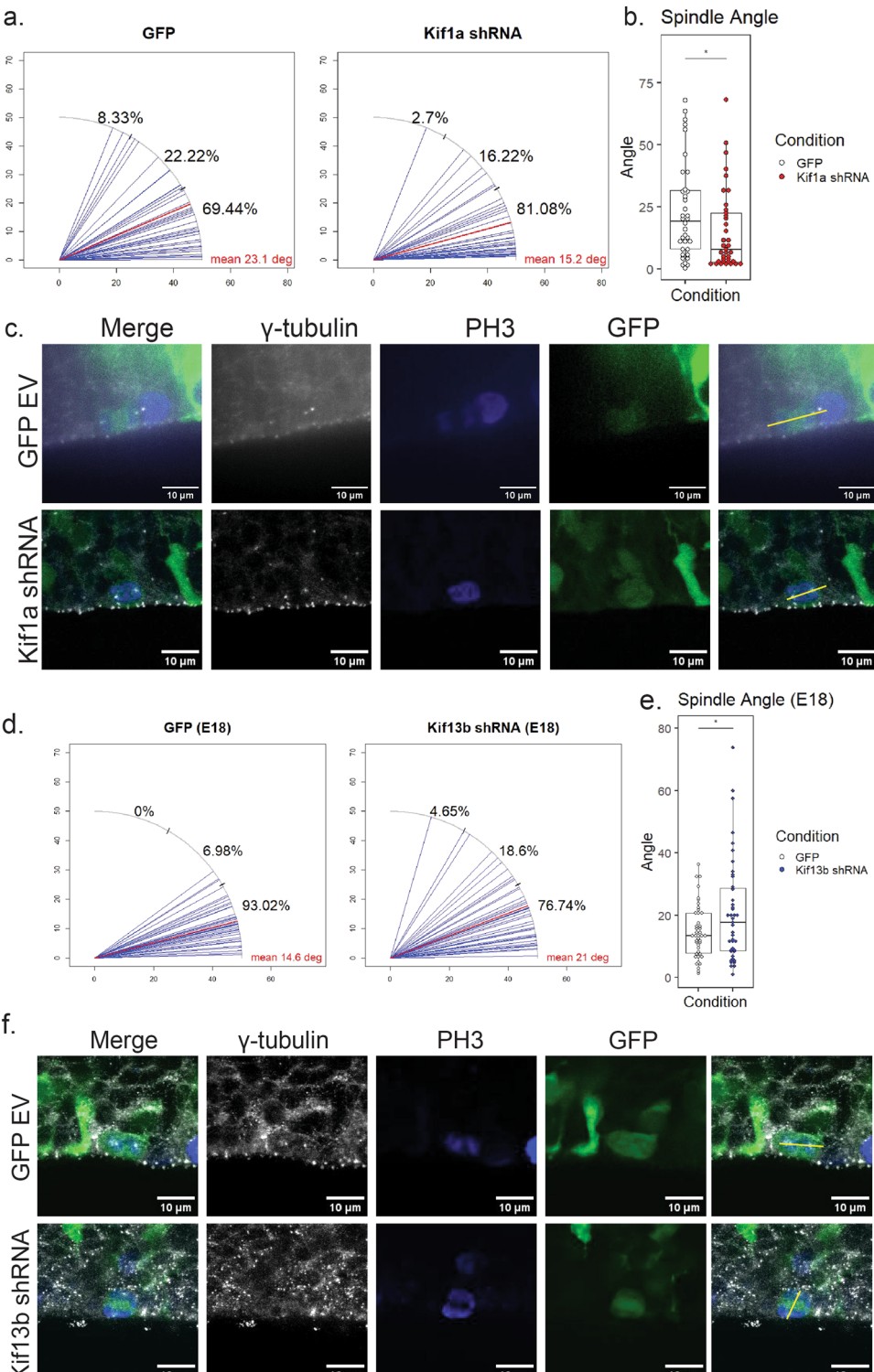

lysate[28]. The main difference between the two kinesins, and in fact a main difference between Kif13B and most kinesins, is the presence of a CAP-Gly domain, which serves to preferentially associate with tyrosinated tubulin, as is seen in astral microtubules[29]. Kif1A, lacking such a domain, has no preference for tyrosinated MTs and actually preferentially disassociates from the GTP-tubulin plus end of MTs[30]. Thus, if both kinesins can bind Dlg1 at the cell membrane during mitosis, Kif13B could remain anchored there to astral MTs through function of its CAP-Gly domain, while Kif1A would disassociate from the plus ends of MTs,

severing the connection between the mitotic spindle and the cell membrane, allowing for more free movement of the spindle. This potential mechanism would require a more detailed analysis of the function of each domain within Kif13B and Kif1A, and their regulation throughout the cell cycle.

The work shown in this study indicates that the regulation of neurogenesis in the mammalian cortex is a highly regulated and complex process. The balance of asymmetric:symmetric divisions must be tightly controlled in order to produce all the neurons required for a functional cortex, but also maintain the population of

**Fig. 5 Kif13b shRNA and Kif1a shRNA have opposite effects on spindle orientation. a** Plots of spindle angle in E19 rat brain in utero electroporated with GFP control (left) and Kif1a shRNA (right) conditions. Mean spindle angle relative to the ventricular surface for each condition is shown in red. Percentages shown in black show percent of cells with spindle angles in 0–30, 30–60, and 60–90 degree range. (Total cells from GFP control = 36 cells from 10 brains, Kif1a shRNA = 37 cells from 5 brains). **b** Plot of spindle angles in GFP (white) and Kif1a shRNA (red) conditions. Data plotted as interquartile range with 5–95% whisker range. (*$p < 0.05$, analyzed using one sided unpaired $t$-test.). **c** Representative image of GFP EV and Kif1a shRNA-expressing RGPs at E19. Gamma tubulin is shown in gray, PH3 in blue, and GFP construct in green. A yellow line indicates the spindle angle relative to the ventricular surface. Scale bar = 10 µm. **d** Plots of individual angles in E18 rat brains in GFP EV (left) and Kif13b shRNA (right) conditions. Mean spindle angle for each condition is shown in red. Percentages shown in black show proportion of cells with mitotic spindles angles relative to the ventricular surface within 0–30, 30–60, or 60–90 degree range. (Total cells from WT control = 43 cells from 9 brains, Kif13b shRNA = 43 cells from 11 brains). **e** Plot of spindle angles in GFP (white) and Kif13b shRNA (blue) conditions. Data plotted as interquartile range with 5–95% whisker range. (*$p < 0.05$, analyzed using one sided unpaired $t$-test). **f** Representative images of mitotic spindles in GFP EV and Kif13b shRNA mouse brains at E18. PH3 is shown in blue, gamma tubulin is shown in gray. A yellow line indicates the angle of each spindle. Scale bar = 10 µm.

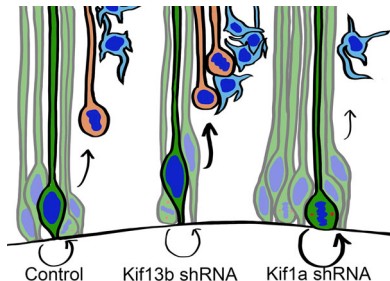

**Fig. 6 Diagrammatic representation of the effect of kinesin shRNA on IP production and neurogenesis.** Under control conditions, the balance between asymmetric and symmetric divisions results in some production of IPs, while maintaining the RGP population. In Kif13b shRNA conditions, more randomized spindle angles result in overproduction of IPs early in development resulting in a burst of neurogenesis. In Kif1a shRNA conditions, a more strictly controlled spindle angle results in reduced IP production and decreased neurogenesis.

RGPs in order to maintain neurogenesis throughout development. Here we have introduced Kif13B as a key regulator of this process through its function in spindle orientation. Kif1A, previously shown to affect the plane at which RGPs divide[5], affects RGP spindle orientation in the opposite manner of Kif13B, which provides a more detailed regulation of RGP cell fate control, through the opposing effects of two very closely related kinesins.

## Methods

All animal experiments were performed under protocols approved by the Institutional Animal Care and Use Committee (IACUC) at Columbia University. We have complied with all relevant ethical regulations for animal use.

**In utero electroporation**. In utero electroporation procedures were performed as follows[31]. Timed pregnant rats were anaesthetized at E16. The abdomen was shaved and sterilized and an incision was made along the midline of the abdomen, about 2–3 centimeters long. Each horn of the uterus was exposed and each embryo was held against a lamp to transilluminate the skull. One to two µl of prepared DNA and surgical dye was injected in the left ventricle with a glass needle. Each embryo was then electroporated with a tweezer electrode. The uterus was returned to the abdomen and the rat was sutured. The rat was anesthetized again at E18, E19, or E20 and embryonic brains were dissected and placed in 4% paraformaldehyde, then incubated at 4 °C shaking for 12–15 h. Fixed brains were embedded in 4% agarose in PBS and sliced into 100 um thick slices and stored at 4 °C. Sliced brains were screened for presence of the electroporated marker and transfected slices were selected for staining. For all conditions, at least three brains from at least two mothers were included in statistical analysis. All rats used were wild-type Sprague Dawley females.

**Mouse procedures**. Kif13B fl/fl mice were a gift from the Bolino lab[11]. Nestin Cre mice were purchased from Jackson Labs (strain: 003771) and mated to Kif13B fl/fl mice. Timed mouse matings were initiated after female mice had been exposed to soiled male bedding for 3 days. On the evening of the third day, one female and one male mouse per cage were co-housed overnight and separated the following morning. The morning of separating was designated as E0.5. At E14.5, female mice were anesthetized with ketamine and xylazine and embryonic brains were recovered and placed in 4% paraformaldehyde, then incubated at 4 °C shaking for 12–15 h. Fixed brains were embedded in 4% agarose in PBS and sliced into 100 um thick slices and stored at 4 °C. All mice genotypes were on a C57BL/6 background.

**Immunostaining**. Screened brain slices were blocked in 5% normal donkey serum (NDS) in PBS with 0.3% Triton X-100 for 1 h at room temperature. Primary antibodies were diluted in PBS with 5% NDS and incubated overnight at 4 °C. Slices were rinsed with PBS, then incubated with secondary antibodies, and DAPI if necessary, for 2 h at room temperature in PBS. Slices were then rinsed in PBS and mounted onto glass slides with Aqua Poly-Mount media and coverslipped.

**Microscopy**. Fixed brain slices were imaged at z-stacks of 2 µm for 10x images, and the ventricular zone of electroporated brains were imaged at 60x in z-stacks of 0.5 µm or 0.75 µm. Images were analyzed using ImageJ and Fluorender software. Spindle angles were identified with ImageJ and Fluorender software and calculated with R code provided in Juschke et al.[32]. ImageJ was used to identify cells positive for both GFP and PH3 in rat brains, or positive for PH3 in mouse brains, then Fluorender was used to identify the three-dimensional coordinates of five points on the ventricular surface, as well as the two centrosomes in the dividing cell. The coordinates for all points were loaded into R and the spindle angle was calculated. All graphs were generated with the R package ggplot2.

**Statistics and reproducibility**. All statistical analysis was performed using R or Prism (GraphPad Software, La Jolla, CA, USA). Data were tested with the D'Agostino-Pearson omnibus normality test, to determine whether they followed a normal distribution. With normal distributions, an unpaired $t$-test was used. When the normality test failed, the non-parametric Mann–Whitney test or one-way ANOVA were used. Significance was accepted at the level of $p < 0.05$.

**Reporting summary**. Further information on research design is available in the Nature Portfolio Reporting Summary linked to this article.

## Data availability

All data supporting the findings of this study are available within the paper and its Supplementary Information. Supplementary Data 1 includes all data used in each figure in this study.

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

## Author contributions

P.H. and R.B.V. conceived the project. P.H. performed all experiments and data analysis. P.H. and R.B.V. wrote the paper.

## Competing interests

The authors declare no competing interests.
