## [Peer Review File · Communications Biology]

Reviewers' comments:

Reviewer #1 (Remarks to the Author):

This manuscript reported novel phenotypes of Kif13b and Kif1a knockdown in rat radial glial progenitor (RGP) cells. They proposed that these two kinesins have opposing roles in neurogenesis via regulating the ratio of symmetric and asymmetric divisions. Overall, the findings are very interesting and will be suitable for general readers of Communications Biology. The major issue of this manuscript is no mechanism is provided for how Kif13b and Kif1a have opposing functions, given that they belong to the same kinesin-3 sub-family. Without molecular mechanisms, the reviewers think it is essential to perform additional experiments to validate their preliminary findings on the phenotypes. Details are given below.

Major comments

1. Kif13b cKO mice were only used for the analysis of mitotic spindle angles, but not earlier analyses on neurogenesis phenotypes. It is important to confirm the neurogenesis defects seen in the knockdown of rat Kif13b using the Kif13b cKO mice in vivo. Conversely, the mitotic spindle orientation phenotypes seen in the Kif13b cKO should be confirmed using Kif13b knockdown in rat RGPs.
2. Similarly, the phenotype for Kif1A KD in neurogenesis and mitotic spindle orientation is only shown in rat RGPs. It would be important to analyze the kif1A KD in developing mouse brains and find out if similar effects are observed for mouse Kif1A on RGP cell fate. Given that Kif1A KD and Kif13b KD showed an opposite effect on mitotic spindle orientation and RGP cell fate determination, it is essential to ensure the opposite effect of both molecules is conserved.
3. There is no gain-of-function analysis for Kif13b and Kif1a in RGPs in this manuscript. Would it generate the opposite effect to the respective KD? This is critical to support the major conclusion of the manuscript.
4. A co-knockdown in RGPs should be performed for kif13b and kif1a. If the phenotypes in neurogenesis are suppressed somewhat back to the wild-type level, it would support their opposing roles.

Minor comments:

1. "Interestingly, the fraction of GFP+/TBR2+ cells was also increased by Kif1a shRNA, with 46% ± 13% of cells positive for GFP and TBR2. As the percent of GFP+/Pax6+ cells is not reduced by Kif1a shRNA treatment, many of these cells are likely to be positive for both Pax6 and TBR2." The quantification on the percent of Pax6 and TBR-double positive, GFP+ cells should be included to support the claim.
2. Even though there is no mechanism given for the opposite effect on mitotic spindle orientation by Kif13b and Kif1a, a discussion on potential candidates, ie common/different downstream cargo proteins, is needed.
3. Poor callout of figures in the manuscript throughout the entire manuscript. Whenever the figure is called out in the manuscript, it should be given the details, such as Fig 1A, instead of Fig 1. Currently, it is hard for the reviewer to match the figure panels with the manuscript text.

Reviewer #3 (Remarks to the Author):

In the article, Helmer and Vallee address the role of the two related kinesins Kif1A and Kif13B in the balance between symmetrical and asymmetrical division of Radial Glial Progenitor Cells (RGPs). They propose that these two kinesins have opposite effects, Kif1A promoting asymmetric division and Kif13B promoting symmetric divisions, through an effect on spindle orientation. Kif13B has never been studied in this system and the shRNA phenotypes on cell fate and cell localization are striking. Some major concerns however must be addressed, in particular whether cell fate defects in Kif13B loss of function are linked to spindle orientation defects, and whether Kif13B and Kif1A really have opposing

effects. Overall, the Kif1A data does not add much to the manuscript and the authors could focus on a careful study of Kif13B.

Major comments:

1/ The authors observe clear cell fate defects in Kif13B shRNA (premature differentiation). However, they do not detect spindle orientation with these constructs and justify using a mouse KO because spindle defects are hard to spot at E19 (already randomized). While the use of a genetic conditional KO is certainly a strong plus in the study, there are a few problems with this reasoning. First and foremost, if the authors do see cell fate defects in the Kif13B shRNA condition and attribute them to spindle defects, they should be able to detect them at early stages (E16 or E17). Without spindle orientation defects in the very condition that shows cell fate defects, the conclusion does not hold. This is critical as a link between spindle orientation and fate remain debated in these cells. The authors can alternatively perform a cell fate analysis in the KO mouse.

2/ Related to this, the authors claim that at E19 the spindle is somewhat randomized. What is the argument for this?

3/ It would have been much better to quantify spindle orientation at anaphase, which is both more precise and what matters in the end. Some images (as in 5F) look more like pro-metaphase, where spindle orientation is still ongoing.

4/ The cell fate analysis fits the authors' model for Kif13B but not for Kif1A. The explanation in the text is not very clear. If Kif1A KD leads to increased symmetric cell division, the fraction of PAX6+ cells should increase, and the number of TBR2+ cells decrease (the opposite is observed). From these data, it is really not clear that the two kinesins have opposing roles on neurogenesis. Figure 1 shows positional differences between Kif13B and Kif1A but some of these could be due to neuronal migration differences.

Minor points

1/ First paragraph of the introduction: define RGP

2/ "interkinetic nuclear migration (INM) a unique feature of neural progenitor development". INM is not unique to these cells.

3/ References of Figure 4 are missing in the text.

4/ The text is sometimes confusing due to the use of both "division plane" or "spindle angle". It would be nice to use only one of the two throughout the manuscript and define it clearly.

Reviewer #4 (Remarks to the Author):

Helmer and Vallee report results from their study of the kinesin Kif13b in the developing mouse brain. Current models suggest division orientation is coupled to fate specification so the mechanisms that might link the two are of great interest. Furthermore, previous work by this group found that the kinesin Kif1a reduces the ratio of asymmetric to symmetric divisions in radial glial progenitor cells and that this effect is correlated with an increase in the division angle and the ventricular surface. In this work, they report that knockdown of Kif13b has the opposite effect. Overall this is an interesting study that implicates another kinesin in mammalian spindle orientation and neurogenesis and it raises intriguing questions about how the kinesin's activity is coordinated to control the balance of

asymmetric and symmetric divisions. However, I do have some concerns about the spindle orientation analysis that should be addressed before publication.

- On line 176, the result of Kif13b shRNA on spindle orientation is referenced but the data aren't shown. Furthermore, the result is described as not statistically significant, but also as an increase, which seems mutually incompatible.
- The comparison of the effects of Kif1a and Kif13b and spindle orientation appears to be problematic as they are very different experiments. Furthermore, the statistical comparisons (i.e. Fig 5B and Fig 5E) should be the same.
- For analyses that show the percentage of cells, the total number of cells should be shown
- There are a number of typos - some additional proofreading is warranted

Reviewer #1 (Remarks to the Author):

This manuscript reported novel phenotypes of Kif13b and Kif1a knockdown in rat radial glial progenitor (RGP) cells. They proposed that these two kinesins have opposing roles in neurogenesis via regulating the ratio of symmetric and asymmetric divisions. Overall, the findings are very interesting and will be suitable for general readers of Communications Biology. The major issue of this manuscript is no mechanism is provided for how Kif13b and Kif1a have opposing functions, given that they belong to the same kinesin-3 sub-family. Without molecular mechanisms, the reviewers think it is essential to perform additional experiments to validate their preliminary findings on the phenotypes. Details are given below.

Major comments

1. Kif13b cKO mice were only used for the analysis of mitotic spindle angles, but not earlier analyses on neurogenesis phenotypes. It is important to confirm the neurogenesis defects seen in the knockdown of rat Kif13b using the Kif13b cKO mice in vivo. Conversely, the mitotic spindle orientation phenotypes seen in the Kif13b cKO should be confirmed using Kif13b knockdown in rat RGPs.

More rigorous analysis of the E18 rat brains is now included in Figure 5, showing the change in spindle orientation at that time. Analysis of the mouse model was also expanded in Figure S2, showing similar changes through staining for Pax6, TBR1, and PH3.

2. Similarly, the phenotype for Kif1A KD in neurogenesis and mitotic spindle orientation is only shown in rat RGPs. It would be important to analyze the kif1A KD in developing mouse brains and find out if similar effects are observed for mouse Kif1A on RGP cell fate. Given that Kif1A KD and Kif13b KD showed an opposite effect on mitotic spindle orientation and RGP cell fate determination, it is essential to ensure the opposite effect of both molecules is conserved.

A Kif1A knockout mouse model is not available. Mouse models that include specific point mutations in the motor domain of Kif1A exist, however without knowledge of the mechanism of Kif1A function in spindle orientation, analysis of a motor domain mutation in Kif1A would be less informative than a conditional knockout mouse.

3. There is no gain-of-function analysis for Kif13b and Kif1a in RGPs in this manuscript. Would it generate the opposite effect to the respective KD? This is critical to support the major conclusion of the manuscript.

Overexpression experiments with Kif1A and Kif13B are now included in the manuscript in Figures S4 and S5. Overexpression of Kif13B showed no effect on Pax6, TBR2, or cortical composition. Kif1A OE showed a small increase of cells in the VZ, but no effects on other cortical regions and no effect on Pax6 and TBR2. Both overexpressions resulted in a reduction of mitotic index, resulting in a very small number of PH3 positive cells, which did not allow for effective analysis of spindle orientation or the ratio of APs to IPs, shown in Figure S5.

4. A co-knockdown in RGPs should be performed for kif13b and kif1a. If the phenotypes in neurogenesis are suppressed somewhat back to the wild-type level, it would support their opposing roles.

Experiments with both Kif1a shRNA and Kif13b shRNA are now included in Figure S3 and S5, which show a return to control levels in TBR2 staining and cortical composition, though a difference in Pax6 staining and in percent of transfected cells in the VZ of the cortex still exist in

the double knockout. Double knockdown also resulted in a reduction of mitotic index, which did not allow for effective analysis of mitotic spindle orientation or the ratio of APs to IPs, shown in Figure S5.

Minor comments:

1. "Interestingly, the fraction of GFP+/TBR2+ cells was also increased by Kif1a shRNA, with $46\% \pm 13\%$ of cells positive for GFP and TBR2. As the percent of GFP+/Pax6+ cells is not reduced by Kif1a shRNA treatment, many of these cells are likely to be positive for both Pax6 and TBR2." The quantification on the percent of Pax6 and TBR-double positive, GFP+ cells should be included to support the claim.

Staining for both Pax6 and TBR2 in the same brain was performed to confirm the increase in the population of Pax6+/TBR2+ cells in the Kif1a shRNA condition, shown in Figure S1.

2. Even though there is no mechanism given for the opposite effect on mitotic spindle orientation by Kif13b and Kif1a, a discussion on potential candidates, ie common/different downstream cargo proteins, is needed.

The hypothesis that Kif1A and Kif13B both interact with Dlg1 at the cell cortex is now included. If Kif13B and Kif1A both interact with Dlg1 at the cell membrane during mitosis, the interaction of Kif1A and Kif13B with astral microtubules could explain the opposing effects seen. Kif13B, containing a CAP-Gly domain, would tether the microtubules to the cell membrane through Dlg1. Kif1A would disassociate from the plus end of the MTs, preventing a link between the MTs and cell membrane through Dlg1, and allowing for freer movement of the mitotic spindle. This explanation appears in the Discussion section on pages 13-14.

3. Poor callout of figures in the manuscript throughout the entire manuscript. Whenever the figure is called out in the manuscript, it should be given the details, such as Fig 1A, instead of Fig 1. Currently, it is hard for the reviewer to match the figure panels with the manuscript text.

These errors have been corrected.

Reviewer #3 (Remarks to the Author):

In the article, Helmer and Vallee address the role of the two related kinesins Kif1A and Kif13B in the balance between symmetrical and asymmetrical division of Radial Glial Progenitor Cells (RGPs). They propose that these two kinesins have opposite effects, Kif1A promoting asymmetric division and Kif13B promoting symmetric divisions, through an effect on spindle orientation. Kif13B has never been studied in this system and the shRNA phenotypes on cell fate and cell localization are striking. Some major concerns however must be addressed, in particular whether cell fate defects in Kif13B loss of function are linked to spindle orientation defects, and whether Kif13B and Kif1A really have opposing effects. Overall, the Kif1A data does not add much to the manuscript and the authors could focus on a careful study of Kif13B.

Major comments:

1/ The authors observe clear cell fate defects in Kif13B shRNA (premature differentiation). However, they do not detect spindle orientation with these constructs and justify using a mouse KO because spindle defects are hard to spot at E19 (already randomized). While the use of a genetic conditional KO is certainly a strong plus in the study, there are a few problems with this reasoning. First and foremost, if the authors do see cell fate defects in the Kif13B shRNA condition and attribute them to spindle defects, they should be able to detect them at early stages (E16 or E17). Without spindle orientation defects in the very condition that shows cell fate defects, the conclusion does not hold. This is critical as a link between spindle orientation and fate remain debated in these cells. The authors can alternatively perform a cell fate analysis in the KO mouse.

E16 or E17 analysis is not possible in the IUE model. Surgeries done before E16 result in a large decrease in embryonic survival, and the shRNA constructs need at least 48 hours in order for depletion of target expression. The earliest stage possible to analyze spindle orientation, E18, has now been included showing the increase in spindle orientation in RGPs by E18, shown in Figure 5.

2/ Related to this, the authors claim that at E19 the spindle is somewhat randomized. What is the argument for this?

The distribution of mitotic spindle angles increases over time during development. This is a studied phenomenon, and previously missing relevant references are now included in the manuscript on page 8 with reference 6: Lancaster, M. A. & Knoblich, J. A. Spindle orientation in mammalian cerebral cortical development. *Curr. Opin. Neurobiol.* **22**, 737–746 (2012).

3/ It would have been much better to quantify spindle orientation at anaphase, which is both more precise and what matters in the end. Some images (as in 5F) look more like pro-metaphase, where spindle orientation is still ongoing.

The representative images have been changed to show cells in anaphase, in Figures 5 and S2.

4/ The cell fate analysis fits the authors' model for Kif13B but not for Kif1A. The explanation in the text is not very clear. If Kif1A KD leads to increased symmetric cell division, the fraction of PAX6+ cells should increase, and the number of TBR2+ cells decrease (the opposite is observed). From these data, it is really not clear that the two kinesins have opposing roles on neurogenesis. Figure 1 shows positional differences between Kif13B and Kif1A but some of these could be due to neuronal migration differences.

There is a slight increase in Pax6 staining in Kif1a shRNA conditions, but not a significant one. The increase in TBR2 staining can be attributed to an increase in the number of cells that are positive for both Pax6 and TBR2, which has been confirmed in the resubmission through double staining for both markers. This combination of markers is indicative of differentiating IPs that will not divide again, agreeing with the reduction in PH3 positive IPs seen in the Kif1a shRNA position, which would result in fewer neurons produced for every asymmetric division as direct neurogenesis leads to one neuron and indirect neurogenesis through IP divisions leads to at least two and up to eight neurons from one RGP division. This model is depicted in Figure 6.

Minor points

1/ First paragraph of the introduction: define RGP

2/ “interkinetic nuclear migration (INM) a unique feature of neural progenitor development”. INM is not unique to these cells.

3/ References of Figure 4 are missing in the text.

The minor errors referenced are now corrected.

4/ The text is sometimes confusing due to the use of both “division plane” or “spindle angle”. It would be nice to use only one of the two throughout the manuscript and define it clearly.

Spindle angle is used when the angle measured is the line through the centrosomes relative to the ventricular surface. Division plane is used when the angle measured is the line where the cell undergoes telekinesis relative to the ventricular surface. These numbers are roughly 90 degrees off from one another. The division plane is included because previous studies of Kif1A involvement in spindle angle was done in live brain slices, where centrosomes were not marked. This explanation is now included in the manuscript on page 7.

Reviewer #4 (Remarks to the Author):

Helmer and Vallee report results from their study of the kinesin Kif13b in the developing mouse brain. Current models suggest division orientation is coupled to fate specification so the mechanisms that might link the two are of great interest. Furthermore, previous work by this group found that the kinesin Kif1a reduces the ratio of asymmetric to symmetric divisions in radial glial progenitor cells and that this effect is correlated with an increase in the division angle and the ventricular surface. In this work, they report that knockdown of Kif13b has the opposite effect. Overall this is an interesting study that implicates another kinesin in mammalian spindle orientation and neurogenesis and it raises intriguing questions about how the kinesin's activity is coordinated to control the balance of asymmetric and symmetric divisions. However, I do have some concerns about the spindle orientation analysis that should be addressed before publication.

- On line 176, the result of Kif13b shRNA on spindle orientation is referenced but the data aren't shown. Furthermore, the result is described as not statistically significant, but also as an increase, which seems mutually incompatible.

More robust analysis of Kif13b shRNA on spindle orientation at E18 is now included, which is significantly different than that of GFP control, and the data referenced is included in the corresponding figure, Figure 5.

- The comparison of the effects of Kif1a and Kif13b and spindle orientation appears to be problematic as they are very different experiments. Furthermore, the statistical comparisons (i.e. Fig 5B and Fig 5E) should be the same.

The analysis is now the same for each condition shown.

- For analyses that show the percentage of cells, the total number of cells should be shown

Total cell numbers in addition to the number of brains analyzed are now included in the description of analysis in the figure legends.

- There are a number of typos - some additional proofreading is warranted

Typos have been corrected wherever found.

REVIEWERS' COMMENTS:

Reviewer #1 (Remarks to the Author):

The authors have done a good job addressing the reviewer's comments. The review now recommend the publication of the revised manuscript.

Reviewer #3 (Remarks to the Author):

The authors have now addressed my comments, and greatly improved the manuscript. The link between the spindle orientation defects and the cell fate defects are now much more convincing. I am happy to recommend publication of the manuscript.